# Dietary Supplementation with Botanical Blends Modified Intestinal Microbiota and Metabolomics of Weaned Pigs Experimentally Infected with Enterotoxigenic *Escherichia coli*

**DOI:** 10.3390/microorganisms11020320

**Published:** 2023-01-27

**Authors:** Cynthia Jinno, Kwangwook Kim, Braden Wong, Emma Wall, Ravichandran Sripathy, Yanhong Liu

**Affiliations:** 1Department of Animal Science, University of California, Davis, CA 95616, USA; 2AVT Natural, Vazhakkulam, Aluva 680017, Kerala, India

**Keywords:** botanical blends, *Escherichia coli* challenge, metabolomics, microbiome, weaned pigs

## Abstract

The objective of this study was to investigate supplementation of botanical blends (BB) comprised of 0.3% capsicum oleoresin and 12% garlic oil on gut microbiota and metabolomic profiles in serum and ileal mucosa of *Escherichia coli* infected pigs. Sixty weaned pigs were assigned to one of five treatments: negative control (CON−), positive control (CON+), dietary supplementation of 100 ppm BB1, 50 or 100 ppm BB2. All pigs, except CON−, were orally inoculated with 10^10^ CFU F18 ETEC/3-mL dose for 3 consecutive days after 7 d adaption. Feces, ileal digesta and cecal content were collected for 16S rRNA amplicon sequencing. Serum and ileal mucosa underwent primary metabolomics analysis. Supplementing 100 ppm BB1 increased (*p* < 0.05) relative abundances of *Enterobacteriaceae* and *Escherichia–Shigella* in ileum, and the relative abundances of Bacteroidota and *Prevotellaceae* in cecum than CON+ on d 5 post-inoculation (PI). Supplementing 100 ppm BB2 upregulated serum pinitol on d 4 PI and serum cholesterol and aminomalonic acids on d 21 PI, while supplementing 50 ppm BB2 reduced asparagine in ileal mucosa on d 5 PI than CON+. Supplementation with botanical blends modulated ileal and cecal microbiota and serum metabolomics profiles in weaned pigs under *Escherichia coli* challenge.

## 1. Introduction

Feed additives are often incorporated into pig feed to improve nutrient digestibility, disease resistance, and performance [1]. In-feed antibiotics were commonly supplemented to newly weaned pigs to prevent diarrhea induced by F18 enterotoxigenic *Escherichia coli* (ETEC) when pigs are under weaning stress [2]. However, gut microbes can develop antibiotics resistance and be excreted in urine or feces, which can then be transmitted toward the human population [3,4]. Alternative practices are currently under demand to prevent pathogenic activities and diarrhea when in-feed antibiotics for growth-promoting purpose are restricted, and common alternatives include acidifiers, pharmacological levels of minerals, probiotics, and phytochemicals [5].

Phytochemicals are found in botanical extracts and they are plant-derived materials that can possess a large variety of biological activities including antimicrobial and anti-inflammatory properties, which may promote intestinal health and performance of newly weaned pigs under stress [6]. Botanical extracts have been shown to elicit antimicrobial activity by directly disrupting bacterial structures in *E. coli* cells [7]. One study has also presumed that botanical extracts can modify gut microbiota in such a way that gut microbes release beneficial metabolites to promote health and increase anti-inflammatory effects [8]. In our previous study, two botanical blends comprising 0.3% capsicum oleoresin and 12% garlic oil extracted from different varieties of garlic were supplemented to newly weaned pigs challenged with ETEC F18. Results of this study indicated that supplementation of botanical blends reduced frequency of diarrhea and enhanced intestinal morphology of weaned pigs compared with control pigs [9]. A reduced systemic inflammation was also observed in pigs fed with 100 mg/kg (ppm) of botanical blend type 2 [9].

However, the effects of these botanical blends on gut microbiota and metabolomic profiles of serum and intestinal mucosa of weaned pigs were not investigated. Gut microbiota plays a crucial role in regulating host health. Successful manipulation of gut microbiota with dietary constituents is likely to enhance diarrhea resistance by utilizing the antimicrobial activity or colonization resistance by gut microbes [10]. The changes in gut microbiota by diets or disease conditions could further impact their metabolites and the host metabolomic profile [11]. Therefore, the objectives of this study were: (1) to characterize the gut microbiota of weaned pigs supplemented with botanical blends under enterotoxigenic ETEC infection and (2) to assess the impacts of botanical blend supplementation on the metabolomic profile of serum and ileal mucosa of weaned pigs.

## 2. Materials and Methods

### 2.1. Animals and Experimental Design

This study was conducted at the University of California, Davis (UC Davis), and the protocol was review and approved by the Institutional Animal Care and Use Committee (IACUC #20809). A total of 60 weaning crossbred pigs (Yorkshire × Landrace, body weight (BW): 7.17 ± 0.97 kg) were selected from the Swine Teaching and Research Center at UC Davis. Piglets and their sows were not vaccinated for ETEC and not supplemented with antibiotics prior to the study. Piglets used in this study were also used in our previous study [9]. Around 21 days (d) old, piglets were weaned from their sows and were housed in individual pens (0.61 × 1.22 m) for 28 days, including 7 days before and 21 days after the first ETEC challenge, at the Cole A facility at UC Davis. Equal number of gilts and barrows were assigned to one of the five treatments in a randomized complete block design with weight within sex, litter as blocks, and pig as experimental unit. With 12 replicates per treatment, the five dietary treatments included: (1) negative control (CON−): basal nursery diets without ETEC challenge; (2) positive control (CON+): basal nursery diets with ETEC challenge; (3) supplementation of 100 ppm of botanical blend (BB) type 1 (BB1) with ETEC challenge; (4) supplementation of 50 ppm of botanical blend type 2 (BB2) with ETEC challenge; and (5) supplementation of 100 ppm of BB2 with ETEC challenge. BB1 and BB2 had similar proprietary formulation of botanical actives, including 0.3% capsicum oleoresin and 12% garlic extracts. Synthetic garlic oil was used in BB1, while garlic oil in BB2 was extracted by subjecting ground garlic bulbs to a steam distillation process. Hydrogenated vegetable oil was used to encapsulate BB1 and BB2. The dosage of BB1 was based on our previous studies, in which capsicum oleoresin and garlic extract were supplemented individually to weaning pigs [12,13]. Limited studies have investigated the optimal dosage to supplement natural garlic extract to weaned pigs; hence, two doses of BB2 were used as treatments in the present study.

A two-phase feeding program was used with d −7 to 7 post-inoculation (PI) as phase I and d 7 to 21 PI as phase II, thus, eight diets were formulated for the study. Spray-dried plasma, high level of zinc oxide, and antibiotics were not included in the diets. All formulated diets met the nutrient requirements of weaned pigs according to NRC, 2012 (Table 1) [14]. All pigs were fed with these experimental diets in a mash form throughout the experiment.

After 7 days of adaptation, pigs in all treatment groups except the negative control were inoculated with 3 mL of F18 ETEC for three consecutive days starting d 0. Each dose was provided at 10^10^ CFU per 3 mL in phosphate buffer saline. The ETEC inoculums were prepared by the Western Institute for Food Safety and Security at UC Davis. The F18 ETEC was isolated from a field disease outbreak by the University of Illinois Veterinary Diagnostic Lab (isolate number: U.IL-VDL #05-27242) and expresses heat-labile toxin, heat-stabile toxins, and Shiga-like toxins. The dosage has been shown to cause mild diarrhea in previous studies [12,15,16]. All detailed procedures of the animal experiment were also described in Wong et al. [9].

### 2.2. Sample Collection

Tail samples were collected from all piglets to test for their susceptibility to F18 ETEC. Tails were genotyped using a method described in Kreuzer et al. [17] and confirmed that all pigs used in this study were susceptible to F18 ETEC. Fresh fecal samples were collected at the beginning of the experiment (d −7), d 0 before ETEC inoculation, and d 5 and 21 PI for fecal microbiota analysis using 16S rRNA amplicon sequencing [18,19]. On d 5 PI, 30 pigs (six pigs per treatment) were euthanized near the peak of infection. The remaining 30 pigs were euthanized on the terminating day of the study, d 21 PI, during the recovery period from ETEC infection. For euthanasia, pigs were first anaesthetized with a 1 mL mixture of telazol (100 mg), ketamine (50 mg), and xylazine (50 mg) by intramuscular injection. Anesthetized pigs were then euthanized with an intracardiac injection of 78 mg sodium pentobarbital (Vortech Pharmaceuticals, Ltd., Dearborn, MI, USA). After euthanasia, ileal digesta and cecal content were collected from all pigs and snap-frozen in liquid nitrogen for gut microbiota analysis.

Excluding pigs in the 100 ppm BB1 group, blood samples were collected from 24 pigs (six pigs per treatment) on d 4 and 21 PI for untargeted metabolomic analysis. Ileal mucosa was collected on d 5 PI and immediately stored in liquid nitrogen for untargeted metabolomic analysis.

### 2.3. Microbiota Analysis

The 16S rRNA amplicon sequencing was used to identify and quantify microbial communities in ileal digesta, cecal content, and fecal samples. Bacterial DNA was extracted from all samples using the Quick-DNA Fecal/Soil Microbe Kit (Zymo Research, Irvine, CA, USA) according to the manufacturer’s instructions. DNA samples were quantified and standardized prior to amplification. Duplicate DNA samples were amplified using PCR of the V4 hypervariable region of the 16S rRNA gene using primers 515F (5′-XXXXXXXXGTGTGCCAGCMGCCGCGGTAA-3′), including an 8 bp barcode (X) unique to each sample followed by a 2 nt Illumina adapter (bold), and 806R (5′-GGACTACHVGGGTWTCTAAT-3′) [20]. Each PCR reaction comprised 2 µL template DNA, 9.5 µL nuclease free water, 12.5 µL GoTaq 2× Master Mix (Promega, Madison, WI, USA), 0.5 µL V4 reverse primer (10 µM), and 0.5 µL barcoded forward primer (10 µM). Amplification was carried out using the following setting: 94 °C for 3 min for initializing denaturation; followed by 35 cycles of 94 °C for 45 s, 50 °C for 1 min, and 72 °C for 1.5 min; and 72 °C for 10 min for final elongation. Agarose gel electrophoresis was used to verify amplicon size for each sample, and amplified samples were then pooled together with the amount of sample added being quantified subjectively based on band brightness in the agarose gel. The pooled sample was then purified using the QIAquick PCR Purification Kit (Qiagen, Hilden, Germany) and submitted to the UC Davis Genome Center DNA Technologies Core for 250 bp paired-end sequencing on the Illumina MiSeq platform (Illumina, Inc., San Diego, CA, USA).

Barcode sequences were removed and the raw fastq files were demultiplexed in sabre (https://github.com/najoshi/sabre (accessed on 29 June 2021). Demultiplexed sequences were imported into Quantitative Insights Into Microbial Ecology 2 (QIIME2; version 2020.8) to remove primers and lower quality reads using the DADA2 plugin [21,22]. Paired-end reads were denoised and merged, and chimeras were removed to construct amplicon sequence variants (ASVs). Representative sequences for each ASV were aligned using MAFFT, and masked alignments were used to generate phylogenetic trees using FastTree2 [23,24]. Python library scikit-learn was used to assign taxonomy based on representative sequences against Silva (version 138), which was pre-trained in QIIME2 to be clipped in to only the V4 hypervariable region and clustered at 99% sequence identity [25,26,27].

### 2.4. Untargeted Metabolomics Analysis

Untargeted metabolomics analysis was performed using gas chromatography (Agilent 6890 gas chromatograph controlled using Leco ChromaTOF software version 2.32, Agilent, Santa Clara, CA, USA) coupled with time-of-flight mass spectrometry (GC/TOF-MS) (Leco Pegasus IV time-of-flight mass spectrometer controlled using Leco ChromaTOF software version 2.32, Leco, Joseph, MI, USA) by the NIH West Coast Metabolomics Center. Metabolite extraction method was derived from a previous study [28]. Approximately 30 μL of serum and 10 mg of ileal mucosa samples were first homogenized using a Retsch ball mill (Retsch, Newtown, PA, USA) for 30 s at 25 times per second. Samples were then vortexed and shaken with an extraction solution pre-chilled at −20 °C, in which the extraction solution consisted of isopropanol, acetonitrile, and water at a ratio 3:3:2 and degassed with liquid nitrogen. Samples were then centrifuged at 12,800× *g* for 2 min to collect the supernatant and divide into two equal aliquots. Aliquots were concentrated at room temperature for 4 h in a cold-trap vacuum concentrator (Labconco Centrivap, Kansas City, MO, USA). Residues were then resuspended in 500 µL of 50% aqueous acetonitrile and centrifuged at 12,800× *g* for 2 min to separate complex lipids and waxes. Resultant supernatant was collected and concentrated in a vacuum compressor. Dried sample extracts were derivatized and mixed with internal retention index markers, fatty acid methyl esters with chain lengths of C8 to C30. Samples were injected for GC/TOF-MS analysis, and all samples were analyzed in a single batch. Data were acquired for MS and mass calibration using FC43 (perfluorotributylamine) prior to analysis sequencing. Metabolite identification was performed based on two parameters: (1) retention index window ± 2000 U (around ± 2 second retention time deviation), and (2) mass spectral similarity plus additional confidence criteria that were based on Fiehn et al. [28].

Raw data were pre-processed directly in Leco ChromaTOF software (v.2.32) for automatic mass spectral deconvolution and peak detection at signal/noise levels of 5:1. The BinBase algorithm was then used to further annotate the peaks within the deconvoluted data [29]. The BinBase algorithm also identified derivatized metabolites by matching the spectral data against the Fiehn mass spectral library and the NIST spectral library based on retention index, validation of unique ions and apex masses, and mass spectrum similarity. InChI key, PubChem ID, and KEGG ID were incorporated to name BinBase compounds. Mass/charge ratio (*m/z*) value of ions in MS was detected.

### 2.5. Statistical Analyses

Sequence files for gut microbiota analysis were exported from QIIME2 and imported into R 4.1.0 for data visualization and statistical analysis (Team, 2021). Shannon and Chao1 indices were measured for alpha diversity by using the estimate_richness function in phyloseq [30]. The Bray–Curtis matrix was used to compare communities’ composition among treatments and days in feces and to compare community among treatments and intestinal segments (ileum vs. cecum). The relative abundance of each taxon in each sample was calculated by dividing the number of taxa by the total number of filtered reads in each sample. All microbiota analyses were performed using the phyloseq package and data were visualized using the ggplot2 package [31]. Normality and homoscedasticity were tested using the Shapiro–Wilk test and Bartlett test, respectively. A linear mixed-effect model was fitted using the lme4 package with treatment and site or day and interaction as fixed effects and pig as random effect [32]. Significance of each term in the model was determined using the F-test as a type 3 analysis of variance using the Anova function in the car package, followed by a group comparison using the cld function in the emmeans package [33,34]. When normality or homoscedasticity was not observed, a nonparametric test was performed using the Kruskal–Wallis sum–rank test in the agricolae package [35]. Bray–Curtis dissimilarity was first tested for homoscedasticity using the betadisper function and confirmed with *p* > 0.05. Statistical significance for beta diversity was then tested using PERMANOVA and the vegan package [36]. Statistical significance was assessed as α = 0.05 and statistical tendency as α = 0.10. The *p*-values were adjusted for multiple comparisons using false discovery rate (FDR).

Metabolomics data were analyzed using different modules of the web-based platform MetaboAnalyst 5.0 (https://www.metaboanalyst.ca; accessed on 15 April 2022) [37]. Peaks were filtered from data. Logarithmic transformation and autoscaling were applied to normalize data. Fold change analysis and t-test were conducted to determine fold change and significance of each identified metabolite. Statistical significance was adjusted with false discovery rate (FDR) with q < 0.2, fold change < 2.0, and variable importance in projection (VIP) score > 1.

## 3. Results

### 3.1. Fecal Microbiota

Within fecal microbiota sequence data, the mean number of reads was 14,530 per sample and the total number of taxa identified was 4,134. Both Shannon and Chao1 indices decreased (*p* < 0.05) in feces as pigs aged from d –7 to 21 PI (Figure 1). On d −7, pigs fed with 50 ppm BB2 had lower (*p* < 0.05) Chao1 index than CON+, otherwise no difference was observed among treatments in both Shannon and Chao1 indices throughout the experiment. The principal coordinate analysis based on Bray–Curtis displayed that the fecal samples collected on d –7 were clustered tightly and away from fecal samples collected on d 0, 5, and 21 PI (Figure 2). Clusters of all treatments were overlapping each other within day on d 0, 5, and 21 PI.

The three most abundant phyla in fecal samples were Firmicutes, Bacteroidota, and Proteobacteria from all treatments throughout the experiment (Table 2). The relative abundance of Bacteroidota, *Bacteroidaceae*, *Muribaculaceae*, *Rikenellaceae*, and *Lactobacillaceae* decreased (*p* < 0.05) through time in fecal samples of pigs. However, the relative abundance of Firmicutes, *Lachnospiraceae*, *Streptococcaceae*, and *Veillonellaceae* increased (*p* < 0.05) when pig age increased. ETEC infection did not affect the relative abundance of Bacteroidota, Firmicutes, and Proteobacteria on d 5 and 21 PI when CON+ was compared with CON−. Supplementation with 100 ppm BB1 or 50 ppm BB2 enhanced (*p* < 0.05) the relative abundance of Bacteroidota (14.96 or 14.04% vs. 7.73%) and Proteobacteria (1.93 or 5.60% vs. 0.34%) but reduced (*p* < 0.05) the relative abundance of Firmicutes (75.28 or 74.27% vs. 84.54%) on d 5 PI, compared with CON−. Supplementation was 100 ppm BB2 also enhanced (*p* < 0.05) the relative abundance of Proteobacteria (5.45% vs. 0.34%) but reduced (*p* < 0.05) the relative abundance of Firmicutes (74.39% vs. 84.54%) on d 5 PI, compared with CON−. At the family level, pigs fed with 50 ppm BB2 reduced (*p* < 0.05) the relative abundance of *Lachnospiraceae* (18.32 vs. 25.87%) on d 5, and pigs fed with 100 ppm BB1 reduced (*p* < 0.05) the relative abundance of *Lachnospiraceae* (9.88 vs. 20.12%) on 21 PI, compared with CON−.

At the genus level, *Lactobacillus*, *Streptococcus,* and *Blautia* were the three most abundant genera in fecal samples throughout the experiment (Table 3). Throughout the experiment, the relative abundance of *Prevotella*, *Agathobacter*, *Blautia*, *Faecalibacterium*, *Lactobacillus*, *Megasphaera*, and *Streptococcus* was increased (*p* < 0.05), but the relative abundance of *Clostridium* sensu stricto and *Lachnoclostridium* decreased in feces through time. ETEC infection reduced (*p* < 0.05) the relative abundance of fecal *Faecalibacterium* (3.21% vs. 5.63%) on d 5 PI and fecal *Prevotella* (0.68% vs. 6.59%) on d 21 PI when CON+ was compared with CON−. Pigs supplemented with 100 ppm BB1 had lower (*p* < 0.05) relative abundance of *Blautia* in feces on d 5 (5.74% vs. 9.19%) and 21 PI (3.13% vs. 7.26%) and had higher (*p* < 0.05) relative abundance of *Escherichia–Shigella* on d 5 PI (1.01% vs. 0.07%), than pigs in CON−. Supplementation with 50 ppm BB2 reduced (*p* < 0.05) the relative abundance of Blautia (5.49% vs. 9.19%) and increased (*p* < 0.05) the relative abundance of *Escherichia–Shigella* (4.52% vs. 0.07%) on d 5 PI compared with CON−.

### 3.2. Intestinal Digesta Microbiota on F18 ETEC Peak Infection

Within ileal digesta and cecal contents of weaned pigs collected on d 5 PI, the mean sampling depth was 21,432 reads and the total number of identified taxa was 2061. In alpha diversity, no difference was observed in both Shannon and Chao1 diversities in cecal contents among treatments (Figure 3). However, CON− was observed to have the highest diversity index in Shannon and Chao1 diversities among all treatment in ileal digesta. In beta diversity, the cluster formed by ileal digesta from CON− was distant from other treatments, while BB clusters were overlapping with each other (Figure 4). In cecal digesta, all treatment clusters overlapped.

In ileal digesta and cecal contents, the three most abundance phyla were Firmicutes, Proteobacteria, and Bacteroidota (Table 4). The relative abundance of Bacteroidota, *Prevotellaceae*, *Lachnospiraceae*, and *Ruminococcaceae* was lower (*p* < 0.05) in the ileum than in the cecum. In ileal digesta, the relative abundance of Bacteroidota and its families *Muribaculaceae* and *Prevotellaceae*, and the relative abundance of Firmicutes families *Ruminococcaceae* and *Selemonadaceae* and Proteobacteria family *Succinivibrionaceae* were lower (*p* < 0.05) in CON+ than in CON−. The relative abundance of *Pasteurellaceae* was greater (*p* < 0.05) in CON+ than in CON−. No difference was observed in cecal content between CON− and CON+. Pigs supplemented with 100 ppm BB1 increased (*p* < 0.05) the relative abundance of *Enterobacteriaceae* (16.25% vs. 0.30%) in the ileum compared with CON+. In cecal content, pigs fed with 100 ppm BB1 had greater (*p* < 0.05) relative abundance of Bacteroidota (15.21% vs. 6.46%) and *Prevotellaceae* (13.26% vs. 5.98%) than CON+, while the relative abundance of *Veillonellaceae* was greater (*p* < 0.05) in CON+ (11.36% vs. 4.32%) than in 100 ppm BB1. The relative abundance of *Enterobacteriaceae* was higher (*p* < 0.05) in cecum of pigs supplemented with 100 ppm BB1, or 50 or 100 ppm BB2 than CON+ (1.93, 3.72, 5.69% vs. 0.07%). No difference was observed in ileal and cecal microbiota composition among BB treatments on d 5 PI.

*Lactobacillus* and *Streptococcus* were the two most abundant genera in ileal digesta and cecal contents (Table 5). Cecal content had greater (*p* < 0.05) relative abundance of *Agathobacter*, *Blautia*, and *Faecalibacterium* than ileal digesta. No difference was observed in the relative abundance of the most abundant genera in the ileum and cecum between CON− and CON+. In ileal digesta, the relative abundance of *Escherichia–Shigella* was greater (*p* < 0.05) in 100 ppm BB1 (16.25% vs. 0.30%) than in CON+. In cecal content, the relative abundance of *Megasphaera* was lower (*p* < 0.05) in 100 ppm BB1 (2.81% vs. 8.22%) than in CON+ and the relative abundance of *Escherichia–Shigella* was greater (*p* < 0.05) when pigs were supplemented with BBs than pigs in CON+.

### 3.3. Intestinal Digesta Microbiota during the Recovery Period of F18 ETEC Infection

The mean number of reads was 24,588 per sample and the total number of identified taxa was 1202 in intestinal digesta from pigs collected on d 21 PI. Supplementing with BB and challenged with F18 ETEC did not affect the Shannon and Chao1 indices in ileal digesta and cecal content (Figure 5). For beta diversity, the 50 ppm BB2 cluster had overlap with the CON– cluster, while the 50 ppm BB2 samples clustered away from CON+ and 100 ppm BB1 clusters in ileal digesta (Figure 6). In cecal content, 100 ppm BB1 was moderately clustered away from the cluster for CON−.

The three most abundant phyla were Firmicutes, Bacteroidota, and Proteobacteria in ileal digesta and cecal content samples from all pigs on d 21 PI (Table 6). The relative abundance of Bacteroidota, *Muribaculaceae*, *Prevotellaceae*, *Lachnospiraceae*, and *Succinivibrionaceae* was lower (*p* < 0.05) in the ileum than in the cecum. F18 ETEC inoculation increased (*p* < 0.05) relative abundance of *Streptococcaceae* (56.81% vs. 1.93%) and *Pasteurellaceae* (8.95% vs. 0.68%) in the ileum when comparing CON+ with CON–. Supplementation with 50 ppm BB2 reduced (*p* < 0.05) the relative abundance of *Streptococcaceae* (15.63% vs. 56.81%) in ileal digesta in comparison to CON+. In cecal content, supplementation with 100 ppm BB2 increased (*p* < 0.05) the relative abundance of *Muribaculaceae* (0.38% vs. 0.25%) when compared with CON+. *Lactobacillus* and *Streptococcus* were the most abundance genera in ileal digesta and cecal content on d 21 PI (Table 7). The relative abundance of *Blautia* was higher (*p* < 0.05), but the relative abundance of *Clostridium* sensu stricto and *Turicibacter* was lower (*p* < 0.05) in cecum than in ileum. The relative abundance of *Strepptococcus* (56.81% vs. 1.93%) in ileal digesta was greater (*p* < 0.05) in CON+ than in CON–.

### 3.4. Metabolomic Profiles

A total of 221 metabolites (117 identified and 104 unidentified) were detected in serum samples on d 4 and 21 PI. VIP scores were computed to assess discriminatory variables in the dataset. On d 4 PI, F18 ETEC infection downregulated methionine, malic acid, galactonic acid, and pinitol, and upregulated oleic acid, arachidonic acid, and lauric acid when CON+ was compared with CON– (Table 8). Supplementation with 100 ppm BB2 upregulated pinitol in comparison with CON+ on d 4 PI. No differential metabolites were identified when pairwise comparing CON+ and 50 ppm BB2, and 50 ppm BB2 vs. 100 ppm BB2 on d 4 PI. On d 21 PI, mannose was downregulated and guanosine and methionine and were upregulated in CON+ in comparison to CON–. Supplementation with 100 ppm BB2 upregulated cholesterol and aminomalonic acid, but downregulated heptanoic acid compared with CON+.

A total of 291 metabolites, including 162 identified and 129 unidentified metabolites were detected in ileal mucosa samples collected on d 5 PI. Asparagine was upregulated in ileal mucosa by supplementing 50 ppm BB2 compared with CON+. No differential metabolites were identified in ileal mucosa when comparing CON– vs. CON+, 50 ppm BB2 vs. CON+, and 50 ppm vs. 100 ppm BB2.

## 4. Discussion

Newly weaned pigs are highly stressed due to sudden dietary and environmental changes, and are more susceptible to ETEC-induced post-weaning diarrhea [38]. The animal trial that was conducted to collect samples analyzed in the present study reported that supplementation of botanical blends could alleviate diarrheal severity and regulate the local and systemic immunity of weaned pigs under ETEC challenge [9]. The two botanical blends used in this experiment comprised 0.3% capsicum oleoresin and 12% garlic oil extracted from different sources. The current research was the follow-up study to investigate the effects of selected botanical blends on gut microbiota and metabolomic profiles in serum and ileal mucosa of weaned pigs infected with ETEC. Results of the present study indicate that ETEC modified the intestinal microbiota and moderately modified the profile of serum metabolites of weaned pigs. Supplementation with botanical blends also influenced intestinal microbiota composition, but their effects on ileal mucosal metabolites were limited.

### 4.1. Fecal Microbiota

In the present study, fecal microbiota was shifted over time throughout the study. Decreased Shannon and Chao1 indices indicate that microbial diversity was reduced due to a decrease in microbial richness in fecal samples. Reduced microbial diversity in fecal microbiota was also reported during the early stage of weaning in pigs [39]. Other studies have reported that microbial diversity in weaned pigs increased through time when feces were sampled at monthly intervals [40,41]. The present study thoroughly investigated the fecal microbiota changes in pigs in negative control by covering the entire post-weaning period with a shorter sampling interval. Our results suggest that the microbial diversity initially decreases soon after weaning, but microbial diversity gradually increases as the pigs mature, likely due to a dietary change from sow milk to plant-based dry feed [42]. Principal coordinate analysis (PCoA) plots in the current study also support that age is likely the main driver for fecal microbiota shifts [43]. The two most abundant phyla in fecal samples of all weaned pigs were Firmicutes and Bacteroidota, which was also consistent with the observations in previous studies [44,45]. Decreased abundance in Bacteroidota over time was observed throughout the experiment in the present study and Ma et al. [46], but Lim et al. [47] reported that the relative abundance of Bacteroidota was increased by age in healthy weaned pigs. The present study also observed an increase in the relative abundance of Firmicutes over time throughout the experiment. Increased abundance of Firmicutes and decreased abundance of Bacteroidota are often observed as a potential indicator for dysbiosis [48]. This result may imply that weaning stress has potentially induced a temporary microbial imbalance during the early stage of weaning, but the result does not explain if weaning stress causes a long-term effect on the gut microbiota.

Fecal microbiota was also modified by the presence of ETEC and dietary supplementation with botanical blends. Supplementation with 50 ppm botanical blend 2 increased the relative abundance of *Lachnoclostridium* in feces on d 5 and 21 PI, compared with the positive control. Previous research reported that *Lachnoclosridium* could produce butyrate, which helps maintain energy homeostasis and stimulates immune responses in the small intestine of pigs [49,50]. The source of garlic oil had limited effects on fecal microbiota, except for the relative abundance of *Lachnospiraceae* that was greater in pigs fed with 100 ppm botanical blend 2 (21.94%) than botanical blend 1 (9.88%) on d 21 PI. Overall, fecal microbiota was mainly affected by age rather than botanical supplementations.

### 4.2. Intestinal Digesta during Peak ETEC Infection

The peak of ETEC infection in post-weaning pigs is approximately day 5 to 7 post-inoculation [38]. In the present study, ETEC infection reduced microbial richness and evenness in ileal digesta of weaned pigs during the peak of ETEC infection. However, no difference in alpha diversity and beta diversity was observed in cecal contents when pigs in the positive control were compared with pigs in the negative control. ETEC colonize in the small intestine, thus these results indicate that ETEC inoculation perturbed the gut microbial community more in the ileum than in the cecum [51]. Our results agree with findings of a previous study that Firmicutes and Proteobacteria are predominantly abundant in ileal digesta of weaned pigs [52]. With reduced microbial richness and evenness in ileal digesta of pigs infected with ETEC, differences in microbial taxa abundance were expected between sham and ETEC infected pigs. ETEC infection reduced the relative abundance of *Ruminococcaceae* and *Prevotellaceeae*, which are associated with producing butyrate and contributing to antimicrobial activity in the intestines [53,54]. Moreover, *Pasteurellaceae* was also more abundant in infected pigs than in sham pigs, which was also observed by Li et al. [55]. This observation is in close agreement with a previous study that reported the increase in *Pasteurellaceae* might be correlated with an increase in ETEC [56]. This result also implied that pigs with ETEC infection are potentially undergoing dysbiosis, as increased *Pasteurellaceae* is an indicator of gut dysbiosis in humans with inflammatory bowel disease [57]. It was expected that greater microbial diversity would be observed in the cecal contents than in ileal digesta because the large intestine is a major site for microbial colonization while the small intestine is mainly responsible for nutrient digestion and absorption. In the present study, we did not observe difference in taxa abundance in cecal contents between the negative control and positive control on d 5 PI. This result suggests that the high microbial diversity in the cecum may increase colonization resistance, which prevents ETEC from colonizing further into the large intestine [58]. In addition, the relatively high amount of short-chain fatty acids produced in the large intestine might be another reason for the increased colonization resistance of ETEC [59].

Supplementation with botanical blends modified the intestinal microbiota of weaned pigs during the peak of ETEC infection. Pigs supplemented with 100 ppm botanical blend 1 had greater abundances of *Enterobacteriaceae* and *Escherichia–Shigella* in ileal digesta than pigs in the positive control. The performance and clinical data from these pigs reported that supplementation with botanical blends reduced diarrheal frequency in weaned pigs infected with ETEC, thus there might be other reasons for the increased abundance of *Enterobacteriaceae* in the ileal digesta of pigs in the botanical blend groups [9]. The relative abundance of *Prevotellaceae* was greater in the cecum when pigs were supplemented with 100 ppm botanical blend 1 than pigs in the positive control. *Prevotella* was likely responsible for the increase in *Prevotellaceae*. Similar results were also observed when growing pigs were supplemented with essential oil blends [60]. The present study also observed an increased abundance of *Megasphaera* in the cecum when pigs were supplemented with 100 ppm botanical blend 1. In consistency, Li et al. [61] reported an increase in *Megasphaera* in the cecal microbiota of weaned pigs when supplemented with 62.5 ppm carvacrol and 7.5 ppm thymol. *Megasphaera* can utilize dietary protein and aid in amino acid metabolism in the small intestine of pigs [62], however, research on the role of *Megasphaera* in the large intestine is limited. In the present study, pigs supplemented with botanical blends had lower abundance of *Veillonellaceae* than control pigs, which was different from the observations in Li et al. [61]. This is likely due to the different compositions of plant extracts and oils that were used in these studies. The impacts of botanical blend vary due to their major active components. The present study also indicates that the source of garlic oil has no effects on intestinal microbiota composition of weaned pigs when challenged with ETEC.

### 4.3. Intestinal Digesta during ETEC Recovery Period

Weaned pigs were reported to fully recover from ETEC infection by d 21 PI, as indicated by the normal diarrhea scores [63]. In the present study, no difference in microbial evenness and richness was observed in the ileum and cecum of pigs during the recovery phase of ETEC infection. The PCoA plot suggests that the overall bacterial community in ileal digesta of weaned pigs was similar between pigs in negative control and pigs fed with 50 ppm BB2, while pigs supplemented with 100 ppm BB1 had a different bacterial community in ileal digesta and cecal content compared with negative control. Looking into the taxonomic abundance, *Streptococcaceae* and *Pasteurellaceae* were more abundant in ileal digesta of pigs in the positive control than negative control. These results may imply that ETEC has a long-term effect on the ileal microbiota of weaned pigs. A high abundance of *Streptococcaceae* is correlated with the high abundance of Streptococcus. Other research also reported that pigs under ETEC challenge had high abundance of *Streptococcus* in their gut microbiota [56]. Pathogenic *Streptococcus* spp. are known to disrupt immunoglobulins from eliminating pathogens that invade the intestines [64]. Since 50 ppm BB2 reduced the abundance of *Streptococcaceae* in ileal digesta of weaned pigs challenged with ETEC, this result might suggest that 50 ppm BB2 may modify the intestinal environment to prevent other opportunistic pathogens from invading the gut. However, the present study did not characterize the gut microbiota to the species level, thus future studies are needed to quantify microbiota at the species level and confirm the potential pathogenicity of *Streptococcus*. In addition, supplementation with 100 ppm botanical blend 2 increased the relative abundance of *Muribaculaceae* in the cecum compared with the positive control. *Muribaculaceae* is often known to be one of the predominant families found in mouse cecal microbiota and their abundance can be altered by diets [65,66]. However, the major function of *Muribaculaceae* in the intestine of pigs is not well understood.

### 4.4. Metabolomic Profile

Untargeted metabolomics was performed in serum samples collected on d 4 and 21 PI to identify metabolomic changes in weaned pigs infected with ETEC. During the acute response of ETEC (d 4 PI), ETEC challenge downregulated pinitol, malic acid, galactonic acid, and methionine. Pinitol, malic acid, and galactonic acid are reported to have anti-inflammatory effects via inhibiting the NF-κB activation pathway, thus, suppressing inflammatory cellular responses [67,68,69]. The reduced methionine might be related to the generation of L-cysteine, one of the important substrates required to synthesize glutathione against increased oxidative stress induced by ETEC infection [70]. In addition, ETEC infection upregulated oleic acid, arachidonic acid, and lauric acid, which have been shown to induce inflammation by activating prostaglandins and indirectly activating the NF-κB pathway [71,72]. The results of the serum metabolomic profile are consistent with the results of serum inflammatory mediators that were published in Wong et al. [9], in which higher concentration of serum TNFα (118.55 vs. 74.30 pg/mL) and haptoglobin (2.47 vs. 1.38 mg/mL) were observed in pigs in positive control vs. negative control. The metabolomic profile data also confirm the ongoing systemic inflammation in the ETEC challenged pigs on d 4 PI.

During the recovery period of ETEC infection, ETEC challenged pigs had lower mannose and higher methionine and guanosine in serum samples. The high mannose concentration is associated with increased mannose glycosylation, which was reported to be negatively correlated with intestinal permeability [73]. On d 21 PI, most of the pigs in the present study were recovered from ETEC infection, as indicated by reduced diarrhea and the absence of β-hemolytic coliforms in feces [9]. Thus, the downregulation of mannose also supports that pigs were undergoing intestinal repair with reduced intestinal permeability. Previous research in rats reported that guanosine could alleviate colonic inflammation during colitis challenge [74]. The increased serum guanosine in ETEC challenged pigs also suggests that the intestinal and systemic inflammation was reduced on d 21 PI [9].

On d 4 PI, upregulation of pinitol was observed in pigs fed with 100 ppm botanical blend 2, compared with the positive control. This observation is consistent with the cytokine data published by Wong et al. [9], in which botanical blend supplementation reduced the concentration of TNFα and haptoglobin in serum compared with the positive control. On d 21 PI, supplementation with 100 ppm botanical blend 2 increased aminomalonic acid in comparison to pigs in the positive control. Aminomalonic acid is utilized for iron metabolism and previous studies have reported that other plant extracts could regulate iron metabolism and reverse oxidative damage caused by pathogens [75,76]. This observation indicates that botanical blend supplementation may speed the recovery from intestinal damage caused by ETEC pathogenicity in weaned pigs.

Limited changes in the metabolomic profile were observed in the ileal mucosa of ETEC infected pigs on d 5 PI. The major finding was that supplementation with 50 ppm of botanical blend 2 reduced serum asparagine compared with the positive control. Asparagine is a metabolite from aspartate metabolism. Large amounts of evidence indicate that aspartate could promote a macrophage-mediated inflammatory response and attenuate intestinal damage caused by endotoxins in weaned pigs [77,78]. No differences were observed in serum and ileal mucosa metabolites in pigs fed with different doses of botanical blend 2, which is consistent with intestinal microbiota data and indicates no dose response was observed in botanical blend 2 regarding intestinal microbiota and metabolomics.

## 5. Conclusions

The present study observed that age was the major factor to shift fecal microbiota, indicated by the increased relative abundance of Firmicutes and decreased relative abundance of Bacteroidota throughout sampling time points. Although ETEC infection and botanical blend supplementation had limited impacts on fecal microbiota, they both affected ileal and cecal microbiota composition in weaned pigs during the peak of ETEC infection. Supplementation of botanical blends increased the relative abundance of *Enterobacteriaceae* in ileum, and *Prevotellaceae* and *Megasphaera* in cecum on d 5 PI. Moreover, both ETEC challenge and 100 ppm botanical blend 2 induced some changes in serum metabolomic profile that might be related to the regulation of systemic inflammation in weaned pigs. Limited differences were observed in intestinal microbiota and metabolomics analysis between the two botanical blends. Taken altogether, the present study provided a wider insight on how botanical blends may have reduced inflammation by altering the serum metabolomic profile while minimally affecting the gut microbiota of weaned pigs infected with ETEC. Manipulation of gut microbiota has been considered as a strategy to alleviate post-weaning diarrhea of weaned pigs, but the botanical blends used in the present study could alleviate inflammatory response with limited modulation to the gut microbiota that was analyzed by 16S rRNA sequencing. The limits of 16S rRNA sequencing have been widely recognized. Thus, future studies are suggested to assess gut bacterial biomass and the functional genomes from the gut microbiota. In addition, future studies are also suggested to assess the metabolomic profiles of intestinal digesta to further investigate the impacts of botanical blends on pig gut microbiota.

## Figures and Tables

**Figure 1 microorganisms-11-00320-f001:**
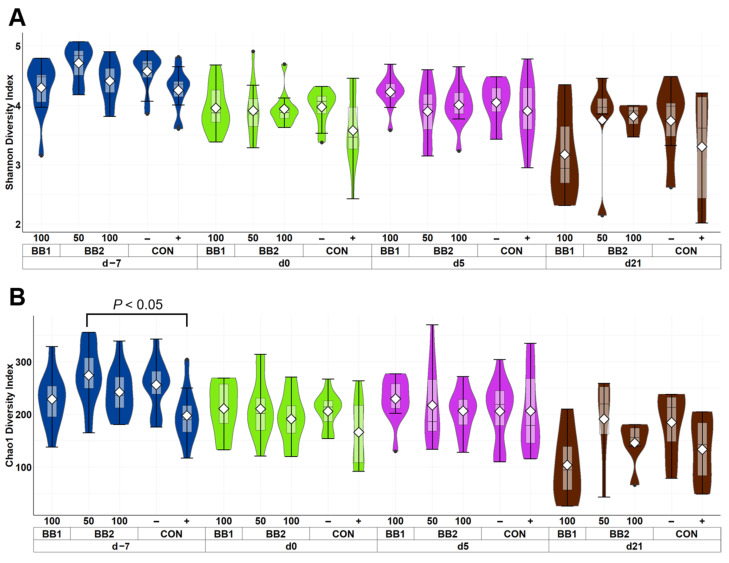
Alpha diversity as indicated by Shannon (**A**) and Chao1 (**B**) indices in feces of weaned pigs fed with control diet (CON), 100 ppm of botanical blend 1 (BB1), or 50 ppm or 100 ppm botanical blend 2 (BB2) at the beginning of the experiment (d −7), first day of F18 ETEC inoculation (d 0), and d 7, 14, and 21 post-inoculation. No difference was observed in Shannon (**A**) index among treatments. Violin plots are colored by day. Data are expressed as mean (diamond) ± SEM. Each treatment had 11–12 observations.

**Figure 2 microorganisms-11-00320-f002:**
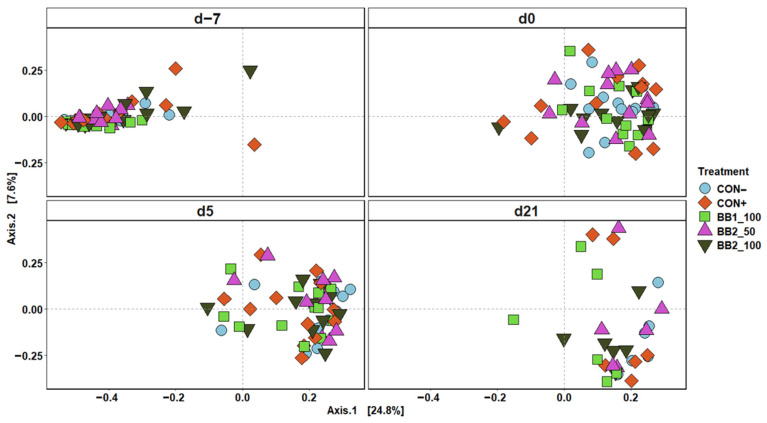
Principal coordinate analysis (PCoA) based on Bray–Curtis distance for beta diversity of fecal samples of weaned pigs fed with a control (CON) diet, or diets supplemented with two different botanical blends (BB). Different colors and shapes represent treatments. The sampling days included d –7 and 0 before ETEC inoculation and d 5 and 21 post-inoculation. CON− = negative control; CON+ = positive control; BB1_100 = 100 ppm BB1; BB2_50 = 50 ppm BB2; BB2_100 = 100 ppm BB2. Each treatment had 11–12 observations.

**Figure 3 microorganisms-11-00320-f003:**
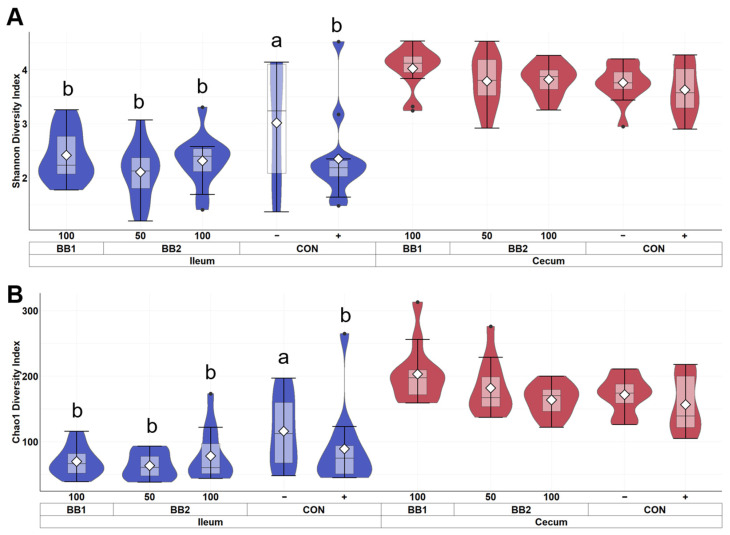
Alpha diversity as indicated by Shannon (**A**) and Chao1 (**B**) indices of ileal and cecal digesta of weaned pigs on d 5 PI fed with control diet (CON), 100 ppm of botanical blend 1 (BB1), or 50 ppm or 100 ppm botanical blend 2 (BB2). Violin plots are colored by site. Data are expressed as mean (diamond) ± SEM. Each treatment had 11–12 observations.

**Figure 4 microorganisms-11-00320-f004:**
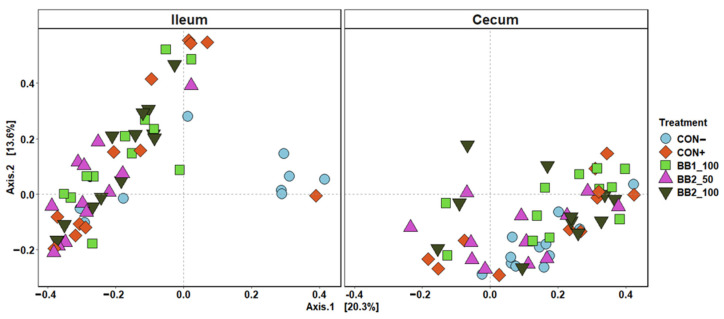
Principal coordinate analysis (PCoA) based on Bray–Curtis distance for beta diversity of ileal and cecal digesta on d 5 post-inoculation of weaned pigs fed with control (CON) diet, or diets supplemented two botanical blends. Different symbols and shapes represent treatments. CON− = negative control; CON+ = positive control; BB1_100 = 100 ppm BB1; BB2_50 = 50 ppm BB2; BB2_100 = 100 ppm BB2. Each treatment had 11–12 observations.

**Figure 5 microorganisms-11-00320-f005:**
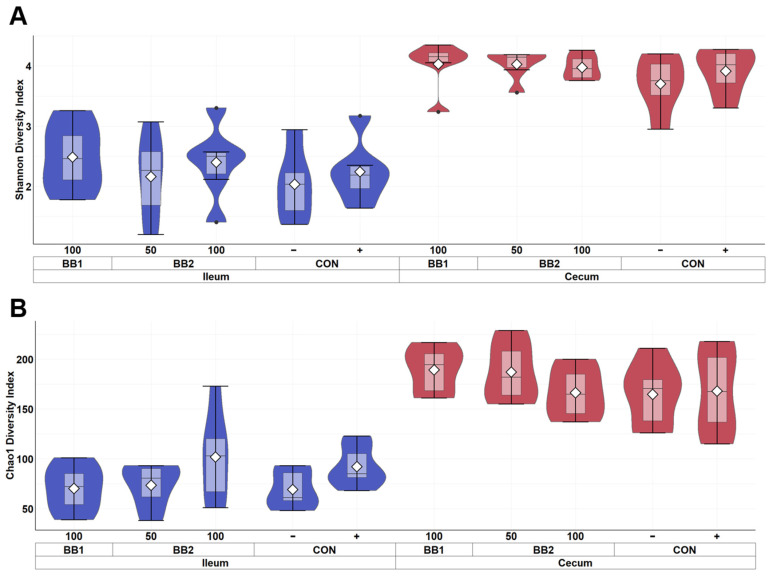
Alpha diversity as indicated by Shannon (**A**) and Chao1 (**B**) indices in ileal and cecal digesta of weaned pigs fed with control diet (CON), 100 ppm of botanical blend 1 (BB1), or 50 ppm or 100 ppm botanical blend 2 (BB2) on d 21 post-inoculation. No difference was observed in Shannon (**A**) and Chao1 (**B**) indices among treatments. Violin plots are colored by site. Data are expressed as mean (diamond) ± SEM. Each treatment had 11–12 observations.

**Figure 6 microorganisms-11-00320-f006:**
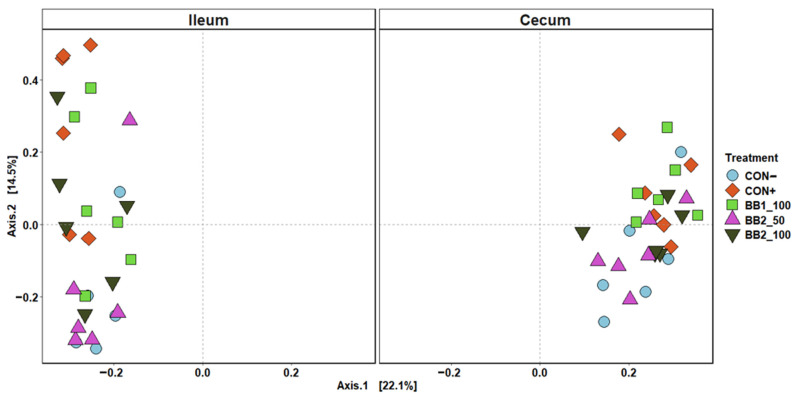
Principal coordinate analysis (PCoA) based on Bray–Curtis distance for beta diversity of ileal and cecal digesta on d 21 PI of weaned pigs fed with a control (CON) diet, or diets supplemented with two botanical blends (BB). Different symbols and shapes represent treatments. CON− = negative control; CON+ = positive control; BB1_100 = 100 ppm BB1; BB2_50 = 50 ppm BB2; BB2_100 = 100 ppm BB2. Each treatment had 11–12 observations.

**Table 1 microorganisms-11-00320-t001:** Ingredient composition of experimental diets, as-fed ^1^.

Ingredient, %	Control, Phase I	Control, Phase II
Corn	51.55	58.44
Dried whey	15.00	10.00
Soybean meal	21.00	24.00
Fish meal	4.00	3.00
Soy protein concentrate	3.00	-
Soybean oil	2.10	1.30
Limestone	0.95	0.95
Dicalcium phosphate	0.55	0.52
_L_-Lysine·HCl	0.48	0.48
_DL_-Methionine	0.24	0.21
_L_-Threonine	0.21	0.20
_L_-Tryptophan	0.09	0.09
_L_-Valine	0.13	0.11
Salt	0.40	0.40
Vitamin–mineral pre-mix ^2^	0.30	0.30
Total	100.00	100.00

^1^ In each phase, three additional diets were formulated by adding 100 ppm of botanical blend type 1, or 50 or 100 ppm of botanical blend type 2 to the control diet, respectively. ^2^ Provided by United Animal Health (Sheridan, IN, USA). The pre-mix provided the following quantities of vitamins and microminerals per kilogram of complete diet: vitamin A as retinyl acetate, 11,136 IU; vitamin D3 as cholecalciferol, 2,208 IU; vitamin E as DL-alpha tocopheryl acetate, 66 IU; vitamin K as menadione dimethylprimidinol bisulfite, 1.42 mg; thiamin as thiamine mononitrate, 0.24 mg; riboflavin, 6.59 mg; pyridoxine as pyridoxine hydrochloride, 0.24 mg; vitamin B12, 0.03 mg; D-pantothenic acid as D-calcium pantothenate, 23.5 mg; niacin, 44.1 mg; folic acid, 1.59 mg; biotin, 0.44 mg; Cu, 20 mg as copper sulfate and copper chloride; Fe, 126 mg as ferrous sulfate; I, 1.26 mg as ethylenediamine dihydriodide; Mn, 60.2 mg as manganese sulfate; Se, 0.3 mg as sodium selenite and selenium yeast; and Zn, 125.1 mg as zinc sulfate.

**Table 2 microorganisms-11-00320-t002:** Relative abundance (%) of the most abundant families from phyla Bacteroidota, Firmicutes, and Proteobacteria in feces of weaned pigs at the beginning of the experiment (d −7), d 0 before first enterotoxigenic *Escherichia coli* inoculation, and d 5, and 21 post-inoculation ^1^.

	d −7					d 0					d 5					d 21				
	Sham	*Escherichia coli* Challenge				Sham	*Escherichia coli* Challenge				Sham	*Escherichia coli* Challenge				Sham	*Escherichia coli* Challenge			
	Negative Control	Positive Control	BB1 100 ppm	BB2 50 ppm	BB2 100 ppm	Negative Control	Positive Control	BB1 100 ppm	BB2 50 ppm	BB2 100 ppm	Negative Control	Positive Control	BB1 100 ppm	BB2 50 ppm	BB2 100 ppm	Negative Control	Positive Control	BB1 100 ppm	BB2 50 ppm	BB2 100 ppm
Bacteroidota	16.51 ^ab^	15.38 ^abcd^	11.68 ^bcde^	19.15 ^a^	15.23 ^abcd^	15.96 ^abc^	10 ^de^	15.30 ^abcd^	12.05 ^bcde^	14.86 ^abcd^	7.73 ^ef^	10.76 ^cde^	14.96 ^abcd^	14.04 ^abcd^	12.39 ^bcde^	9.40 ^def^	2.18 ^f^	6.10 ^ef^	7.48 ^ef^	6.25 ^ef^
*Bacteroidaceae*	1.66 ^a^	2.82 ^a^	4.31 ^a^	2.90 ^a^	3.01 ^a^	0.06 ^bcd^	0.58 ^b^	0.31 ^b^	0.21 ^b^	0.17 ^b^	0.02 ^cd^	0.06 ^bcd^	0.11 ^bc^	0.24 ^bcd^	1.24 ^bc^	0 ^cd^	0 ^d^	0 ^d^	0 ^d^	0 ^d^
*Muribaculaceae*	2.99 ^a^	1.90 ^abc^	0.86 ^c^	2.30 ^abc^	1.79 ^abc^	2.52 ^ab^	1.34 ^bc^	2.64 ^abc^	2.23 ^abc^	2.58 ^abc^	1.72 ^abc^	1.4 ^abc^	2.42 ^abc^	1.39 ^abc^	1.56 ^abc^	0.60 ^c^	0.76 ^c^	0.88 ^c^	1.84 ^abc^	1.84 ^abc^
*Prevotellaceae*	6.96 ^ab^	7.21 ^abc^	3.82 ^bc^	9.39 ^a^	6.38 ^abc^	10.93 ^a^	7.31 ^ab^	9.64 ^a^	8.13 ^ab^	10.30 ^a^	4.91 ^abc^	7.99 ^abc^	10.18 ^a^	10.62 ^a^	8.18 ^ab^	8.51 ^ab^	1.25 ^c^	4.91 ^abc^	4.97 ^abc^	4.22 ^abc^
*Rikenellaceae*	2.15 ^a^	1.97 ^abc^	1.73 ^abcd^	1.95 ^abc^	1.99 ^ab^	1.76 ^abcd^	0.53 ^ghij^	2.25 ^bcdef^	1.08 ^cdef^	1.47 ^abcde^	0.78 ^efghi^	1.06 ^defgh^	1.40 ^abcdef^	1.17 ^defgh^	0.99 ^defg^	0.24 ^hij^	0.10 ^j^	0.21 ^ij^	0.63 ^fghij^	0.15 ^ij^
Firmicutes	66.09 ^hi^	68.65 ^fghi^	71.11 ^efghi^	62.97 ^i^	67.68 ^ghi^	73.69 ^defgh^	75.24 ^cdef^	70.30 ^efghi^	71.98 ^efgh^	71.83 ^efgh^	84.54 ^ab^	79.85 ^bcde^	75.28 ^cdef^	74.27 ^cdefg^	74.39 ^cdef^	83.84 ^abcd^	91.05 ^a^	75.67 ^bcdef^	84.99 ^abc^	82.59 ^abcd^
* Lachnospiraceae*	10.29 ^efg^	9 ^efg^	8.06 ^g^	10.44 ^efg^	8.59 ^fg^	20.29 ^abc^	14.83 ^cdef^	18.13 ^bcd^	18.16 ^bcd^	18.98 ^bcd^	25.87 ^a^	22.84 ^ab^	23.07 ^ab^	18.32 ^bc^	21.77 ^abc^	20.12 ^abc^	14.88 ^bcde^	9.88 ^defg^	15.03 ^bcdef^	21.94 ^abc^
*Lactobacillaceae*	24.71 ^a^	31.10 ^a^	23.32 ^a^	27.49 ^a^	23.22 ^a^	17.75 ^abc^	31.72 ^abc^	23.14 ^abc^	26.67 ^abc^	16.41 ^abcd^	22.35 ^ab^	20.84 ^ab^	19.20 ^ab^	26.35 ^a^	21.29 ^ab^	5.16 ^de^	11.51 ^bcd^	2.46 ^e^	5.15 ^de^	9.93 ^cde^
*Ruminococcaceae*	5.93	5.21	6.85	5.34	4.87	7.84	5.14	7.02	5.54	5.88	10.06	5.70	7.30	6.65	7.78	9.02	8.39	6.54	7.30	11.52
*Streptococcaceae*	0.95 ^bcde^	0.87 ^cdef^	0.60 ^cdef^	0.62 ^cdef^	0.49 ^def^	0.22 ^f^	5.71 ^abcde^	2 ^ef^	0.46 ^def^	2.12 ^cdef^	1.17 ^cdef^	4.16 ^cdef^	3.37 ^abcde^	4.93 ^cdef^	3.72 ^abcde^	12.90 ^abcde^	13.19 ^ab^	19.69 ^a^	12.79 ^abc^	7.14 ^abcd^
*Veillonellaceae*	1.20 ^d^	0.85 ^d^	0.55 ^d^	0.39 ^d^	0.95 ^d^	3.73 ^bc^	5.42 ^abc^	5.78 ^abc^	6.49 ^abc^	5.80 ^abc^	6.30 ^abc^	7.12 ^abc^	4.25 ^c^	6.52 ^abc^	4.87 ^abc^	10.96 ^a^	10.23 ^a^	5.74 ^abc^	7.58 ^abc^	9.40 ^ab^
Proteobacteria	2.36 ^abcd^	1.60 ^abcd^	2.03 ^abcd^	2.18 ^abcd^	1.85 ^abcd^	1.97 ^abcd^	7.26 ^a^	6.39 ^abcd^	7.77 ^ab^	3.36 ^abcd^	0.34 ^e^	1.61 ^cde^	1.93 ^bcd^	5.60 ^abcd^	5.45 ^abcd^	1.04 ^cde^	0.74 ^de^	10.82 ^abc^	0.94 ^cde^	4.05 ^abcd^
*Enterobacteriaceae*	0.75 ^bcd^	0.87 ^bcd^	1.16 ^abc^	0.49 ^bcd^	0.79 ^bcd^	0.46 ^abc^	4.32 ^a^	3.84 ^ab^	4.32 ^a^	2.23 ^abc^	0.07 ^d^	0.64 ^bcd^	1.01 ^abc^	4.52 ^abc^	4.75 ^abc^	0.07 ^cd^	0.13 ^bcd^	1.78 ^abc^	0.37 ^bcd^	1.42 ^bcd^
*Succinivibrionaceae*	1.22 ^ab^	0.44 ^ab^	0.70 ^ab^	1.27 ^ab^	0.68 ^ab^	1.34 ^a^	1.32 ^ab^	0.79 ^ab^	2.44 ^a^	0.78 ^ab^	0.25 ^b^	0.31 ^b^	0.34 ^ab^	0.60 ^ab^	0.30 ^b^	0.96 ^ab^	0.59 ^ab^	2.48 ^ab^	0.41 ^ab^	2.57 ^ab^

^1^ BB1: botanical blend 1; BB2: botanical blend 2. Each treatment had 11–12 observations. ^a–j^ Means without a common superscript are different (*p* < 0.05).

**Table 3 microorganisms-11-00320-t003:** Relative abundance (%) of families in the three most abundant phyla in feces of weaned pigs at the beginning of the experiment (d −7), d 0 before first enterotoxigenic *Escherichia coli* inoculation, and d 5, and 21 post-inoculation ^1^.

	d −7					d 0					d 5					d 21				
	Sham	*Escherichia coli* Challenge				Sham	*Escherichia coli* Challenge				Sham	*Escherichia coli* Challenge				Sham	*Escherichia coli* Challenge			
	Negative Control	Positive Control	BB1 100 ppm	BB2 50 ppm	BB2 100 ppm	Negative Control	Positive Control	BB1 100 ppm	BB2 50 ppm	BB2 100 ppm	Negative Control	Positive Control	BB1 100 ppm	BB2 50 ppm	BB2 100 ppm	Negative Control	Positive Control	BB1 100 ppm	BB2 50 ppm	BB2 100 ppm
Bacteroidota																				
*Prevotella*	2.70 ^ab^	3.20 ^ab^	1.64 ^ab^	3.05 ^ab^	2.60 ^ab^	6.75 ^a^	4.79 ^ab^	5.69 ^a^	3.88 ^ab^	5.42 ^a^	3.07 ^ab^	4.88 ^ab^	5.04 ^ab^	5.20 ^ab^	4.10 ^ab^	6.59 ^a^	0.68 ^b^	3.54 ^ab^	3.51 ^ab^	2.81 ^ab^
Firmicutes																				
*Agathobacter*	0.05 ^e^	0.27 ^cde^	0 ^e^	0.04 ^e^	0.03 ^de^	2.13 ^a^	1.94 ^ab^	1.79 ^a^	3.11 ^a^	3.33 ^a^	2.21 ^a^	2.44 ^a^	2.10 ^a^	1.79 ^ab^	3.37 ^a^	2.37 ^ab^	1.68 ^abcd^	0.53 ^bcde^	1.82 ^abc^	1.47 ^ab^
*Blautia*	0.29 ^e^	0.34 ^e^	0.21 ^e^	0.14 ^e^	0.60 ^de^	7.38 ^ab^	4.51 ^bc^	7.02 ^ab^	4.56 ^bc^	7.45 ^ab^	9.19 ^a^	6.66 ^abc^	5.74 ^bc^	5.49 ^bc^	6.67 ^ab^	7.26 ^ab^	4.31 ^bc^	3.13 ^cd^	4.87 ^bc^	6.37 ^abc^
*Clostridium* sensu stricto 1	7.33 ^a^	6.51 ^a^	10.45 ^a^	6.72 ^a^	6.86 ^a^	0.06 ^b^	0.11 ^b^	0.23 ^b^	0.05 ^b^	0.26 ^b^	0.02 ^b^	0.02 ^b^	0.12 ^b^	0.08 ^b^	0.16 ^b^	0.05 ^b^	1.04 ^b^	0.15 ^b^	0.14 ^b^	0.86 ^b^
*Faecalibacterium*	0.03 ^d^	0.20 ^d^	0.38 ^d^	0.04 ^d^	0.18 ^d^	4.19 ^ab^	1.70 ^c^	3.68 ^abc^	2.75 ^bc^	3.13 ^bc^	5.63 ^a^	3.21 ^bc^	3.33 ^abc^	3.37 ^ab^	4.52 ^ab^	4.27 ^ab^	3.72 ^bc^	3.17 ^bc^	2.86 ^bc^	4.24 ^abc^
*Lachnoclostridium*	5.59 ^a^	5.31 ^a^	5.01 ^a^	6.37 ^a^	4.17 ^a^	0.31 ^bcde^	0.96 ^bcd^	0.50 ^cde^	1.05 ^b^	0.43 ^bcd^	0.88 ^cde^	0.58 ^bc^	0.33 ^bcde^	0.19 ^cde^	0.24 ^bcde^	0.21 ^bcde^	0.12 ^de^	0.10 ^e^	0.11 ^de^	0.21 ^bcde^
*Lactobacillus*	5.16 ^de^	11.51 ^bcd^	2.46 ^e^	5.15 ^de^	9.93 ^cde^	24.71 ^a^	31.1 ^a^	23.32 ^a^	27.49 ^a^	23.22 ^a^	22.35 ^ab^	20.84 ^ab^	19.20 ^ab^	26.35 ^a^	21.29 ^ab^	17.75 ^abc^	31.72 ^abc^	23.14 ^abc^	26.67 ^abc^	16.41 ^abcd^
*Megasphaera*	1.16 ^b^	0.75 ^b^	0.50 ^b^	0.35 ^b^	0.92 ^b^	3.51 ^a^	4.50 ^a^	5.18 ^a^	5.65 ^a^	5.15 ^a^	4.79 ^a^	6.15 ^a^	3.71 ^a^	5.71 ^a^	4.07 ^a^	7.10 ^a^	5.54 ^a^	4.13 ^a^	5.23 ^a^	4.01 ^a^
*Streptococcus*	0.95 ^bcde^	0.87 ^cdef^	0.60 ^cdef^	0.62 ^cdef^	0.49 ^def^	0.22 ^f^	5.71 ^abcde^	2.00 ^ef^	0.46 ^def^	2.12 ^cdef^	1.17 ^cdef^	4.16 ^cdef^	3.37 ^abcde^	4.93 ^cdef^	3.72 ^abcde^	12.9 ^abcde^	13.19 ^ab^	19.68 ^a^	12.79 ^abc^	7.14 ^abcd^
Proteobacteria																				
*Escherichia–Shigella*	0.75 ^bcd^	0.87 ^bcd^	1.16 ^abc^	0.49 ^bcd^	0.79 ^bcd^	0.46 ^abc^	4.32 ^a^	3.84 ^ab^	4.32 ^a^	2.23 ^abc^	0.07 ^d^	0.64 ^bcd^	1.01 ^abc^	4.52 ^abc^	4.75 ^abc^	0.07 ^cd^	0.13 ^bcd^	1.78 ^abc^	0.37 ^bcd^	1.42 ^bcd^

^1^ BB1: botanical blend 1; BB2: botanical blend 2. Each treatment had 11–12 observations. ^a–f^ Means without a common superscript are different (*p* < 0.05).

**Table 4 microorganisms-11-00320-t004:** Relative abundance (%) of Bacteroidota, Firmicutes, and Proteobacteria and their top families in digesta on d 5 post-inoculation of F18 enterotoxigenic *Escherichia coli* infected weaned pigs ^1^.

	Ileum					Cecum				
	Sham	*Escherichia coli* Challenge				Sham	*Escherichia coli* Challenge			
	Negative Control	Positive Control	BB1 100 ppm	BB2 50 ppm	BB2 100 ppm	Negative Control	Positive Control	BB1 100 ppm	BB2 50 ppm	BB2 100 ppm
Bacteroidota	7.13 ^c^	0.78 ^d^	1.41 ^d^	0.14 ^d^	0.36 ^d^	9.96 ^ab^	6.46 ^bc^	15.21 ^a^	10.81 ^ab^	12.3 ^ab^
*Muribaculaceae*	0.51 ^b^	0.17 ^c^	0.02 ^c^	0 ^c^	0.01 ^c^	0.31 ^ab^	0.30 ^ab^	0.52 ^a^	0.39 ^ab^	0.26 ^ab^
*Prevotellaceae*	6.49 ^c^	0.57 ^d^	1.38 ^d^	0.13 ^d^	0.33 ^d^	9.42 ^ab^	5.98 ^bc^	13.26 ^a^	9.64 ^ab^	10.92 ^ab^
Firmicutes	83.77 ^a^	86.22 ^a^	67.26 ^a^	74.48 ^a^	74.25 ^a^	81.98 ^a^	84.98 ^a^	71.72 ^a^	76.77 ^a^	70.90 ^a^
*Lachnospiraceae*	7.34 ^b^	2.97 ^bc^	0.41 ^c^	0.50 ^bc^	0.61 ^bc^	20.43 ^a^	16.32 ^a^	18.54 ^a^	16.32 ^a^	17.33 ^a^
*Lactobacillaceae*	29.35	44.09	37.91	51.18	43.01	27.69	27.22	18.94	29.39	19.91
*Ruminococcaceae*	3.87 ^b^	1.08 ^c^	0.07 ^c^	0.12 ^c^	0.08 ^c^	10.20 ^a^	9.33 ^a^	9.08 ^a^	9.09 ^a^	8.89 ^a^
*Selemonadaceae*	6.23 ^a^	1.16 ^b^	1.45 ^b^	2.34 ^b^	0.79 ^b^	4.81 ^a^	5.78 ^a^	2.97 ^a^	3.29 ^a^	4.82 ^a^
*Streptococcaceae*	6.94	31.07	13.07	12.94	14.24	5.18	8.50	8.13	5.03	3.96
*Veillonellaceae*	8.61 ^abcd^	2.48 ^de^	2.78 ^e^	1.99 ^e^	3.31 ^cde^	6.67 ^abc^	11.36 ^a^	4.32 ^bcde^	7.53 ^ab^	7.34 ^ab^
Proteobacteria	3.95 ^ab^	6.10 ^ab^	20.10 ^a^	17.69 ^ab^	18.2 ^ab^	1.83 ^b^	2.74 ^ab^	4.99 ^ab^	6.06 ^ab^	8.68 ^ab^
*Enterobacteriaceae*	1.91 ^bcd^	0.30 ^bcd^	16.25 ^a^	16.36 ^ab^	16.22 ^abc^	0.43 ^cd^	0.07 ^d^	1.93 ^abc^	3.72 ^abc^	5.69 ^abc^
*Pasteurellaceae*	0.34 ^d^	5.36 ^ab^	3.77 ^a^	1.30 ^abc^	1.75 ^ab^	0.01 ^d^	0.20 ^bcd^	0.20 ^abcd^	0.18 ^cd^	0.07 ^bcd^
*Succinivibrionaceae*	1.64 ^a^	0.13 ^b^	0.04 ^b^	0.01 ^b^	0.01 ^b^	1.33 ^a^	1.74 ^a^	1.91 ^a^	1.32 ^a^	2.84 ^a^

^1^ BB1: botanical blend 1; BB2: botanical blend 2. Each treatment had 11–12 observations. ^a–e^ Means without a common superscript are different (*p* < 0.05).

**Table 5 microorganisms-11-00320-t005:** Relative abundance (%) of the most abundant families in the three most abundant phyla in digesta on d 5 post-inoculation of F18 enterotoxigenic *Escherichia coli* infected weaned pigs ^1^.

	Ileum					Cecum				
	Sham	*Escherichia coli* Challenge				Sham	*Escherichia coli* Challenge			
	Negative Control	Positive Control	BB1 100 ppm	BB2 50 ppm	BB2 100 ppm	Negative Control	Positive Control	BB1 100 ppm	BB2 50 ppm	BB2 100 ppm
Actinobacteria										
*Bifidobacterium*	1.91	4.39	6.47	5.74	4.89	0.72	1.64	1.21	1.61	2.56
Firmicutes										
*Agathobacter*	0.76 ^b^	0.28 ^b^	0.04 ^b^	0.02 ^b^	0.03 ^b^	3.08 ^a^	2.94 ^a^	2.92 ^a^	2.59 ^a^	3.16 ^a^
*Blautia*	1.6 ^b^	1.18 ^bc^	0.05 ^c^	0.09 ^c^	0.09 ^bc^	7.58 ^a^	4.38 ^a^	5.41 ^a^	5.25 ^a^	5.7 ^a^
*Faecalibacterium*	0.89 ^b^	0.46 ^bc^	0.02 ^c^	0.03 ^c^	0.03 ^c^	6.36 ^a^	5.15 ^a^	3.92 ^a^	5.07 ^a^	4.16 ^a^
*Lactobacillus*	29.34	44.09	37.91	51.18	43	27.69	27.22	18.94	29.39	19.91
*Megasphaera*	5.04 ^bc^	1.98 ^c^	2.33 ^c^	1.69 ^c^	2.82 ^bc^	4.1 ^ab^	8.22 ^a^	2.81 ^bc^	5.46 ^ab^	4.71 ^ab^
*Streptococcus*	6.94	31.06	13.07	12.94	14.24	5.18	8.5	8.13	5.03	3.96
*Turicibacter*	6.02 ^abc^	0.33 ^a^	5.11 ^a^	1.15 ^a^	3.34 ^ab^	0.11 ^abcd^	0.05 ^bcd^	0.09 ^cd^	0.07 ^d^	0.25 ^bcd^
Proteobacteria										
*Escherichia–Shigella*	1.91 ^bcd^	0.3 ^bcd^	16.25 ^a^	16.36 ^ab^	16.22 ^abc^	0.43 ^cd^	0.07 ^d^	1.93 ^abc^	3.72 ^abc^	5.69 ^abc^

^1^ BB1: botanical blend 1; BB2: botanical blend 2. Each treatment had 11–12 observations. ^a–d^ Means without a common superscript are different (*p* < 0.05).

**Table 6 microorganisms-11-00320-t006:** Relative abundance (%) of Bacteroidota, Firmicutes, and Proteobacteria and their top families in digesta on d 21 post-inoculation of F18 enterotoxigenic *Escherichia coli* infected weaned pigs ^1^.

	Ileum					Cecum				
	Sham	*Escherichia coli* Challenge				Sham	*Escherichia coli* Challenge			
	Negative Control	Positive Control	BB1 100 ppm	BB2 50 ppm	BB2 100 ppm	Negative Control	Positive Control	BB1 100 ppm	BB2 50 ppm	BB2 100 ppm
Bacteroidota	0.33 ^b^	0.05 ^b^	0.14 ^b^	0.24 ^b^	0.07 ^b^	12.39 ^a^	8.80 ^a^	16.59 ^a^	12.34 ^a^	15.66 ^a^
*Muribaculaceae*	0 ^c^	0 ^c^	0 ^c^	0 ^c^	0 ^c^	0.24 ^ab^	0.25 ^b^	0.27 ^ab^	0.25 ^ab^	0.38 ^a^
*Prevotellaceae*	0.33 ^b^	0.05 ^b^	0.14 ^b^	0.24 ^b^	0.06 ^b^	12.05 ^a^	8.46 ^a^	16.22 ^a^	11.99 ^a^	15.14 ^a^
Firmicutes	91.34	82.67	72.98	74.31	75.81	78.89	82.08	72.11	77.79	69.68
*Lachnospiraceae*	0.16 ^b^	0.52 ^b^	0.38 ^b^	0.63 ^b^	0.85 ^b^	16.18 ^a^	17.20 ^a^	18.87 ^a^	15.39 ^a^	16.19 ^a^
*Lactobacillaceae*	48.62	17.11	36.16	43.76	31.31	27.88	12.68	14.10	24.25	14.22
*Streptococcaceae*	1.93 ^b^	56.81 ^a^	17.83 ^ab^	15.63 ^b^	17.98 ^ab^	8.66 ^b^	14.94 ^ab^	13.9 ^ab^	7.34 ^b^	4.75 ^b^
Proteobacteria	4.50	9.89	20.09	16.24	15.24	2.89	3.71	5.33	3.51	7.51
*Enterobacteriaceae*	3.71 ^ab^	0.30 ^ab^	13.74 ^a^	15.61 ^ab^	12.44 ^ab^	0.43 ^ab^	0.07 ^b^	1.20 ^ab^	1.69 ^ab^	2.27 ^ab^
*Pasteurellaceae*	0.68 ^cde^	8.95 ^a^	6.30 ^ab^	0.58 ^abcd^	2.40 ^abc^	0 ^e^	0.36 ^bcde^	0.29 ^cde^	0.09 ^de^	0.03 ^de^
*Succinivibrionaceae*	0.02 ^b^	0.02 ^b^	0.01 ^b^	0.01 ^b^	0.02 ^b^	2.42 ^a^	3.14 ^a^	3.38 ^a^	1.67 ^a^	5.15 ^a^

^1^ BB1: botanical blend 1; BB2: botanical blend 2. Each treatment had 11–12 observations. ^a–e^ Means without a common superscript are different (*p* < 0.05).

**Table 7 microorganisms-11-00320-t007:** Relative abundance (%) of the most abundant genera from Actinobacteria, Firmicutes and Proteobacteria in digesta on d 21 post-inoculation of F18 enterotoxigenic *Escherichia coli* infected weaned pigs ^1^.

	Ileum					Cecum				
	Sham	*Escherichia coli* Challenge				Sham	*Escherichia coli* Challenge			
	Negative Control	Positive Control	BB1 100 ppm	BB2 50 ppm	BB2 100 ppm	Negative Control	Positive Control	BB1 100 ppm	BB2 50 ppm	BB2 100 ppm
Actinobacteria										
*Bifidobacterium*	3.2	3.85	4.77	6.93	5.98	0.39	0.62	0.26	1.53	1.13
Firmicutes										
*Blautia*	0.05 ^b^	0.08 ^b^	0.05 ^b^	0.13 ^b^	0.12 ^b^	6 ^a^	4.99 ^a^	5.3 ^a^	4.65 ^a^	5.3 ^a^
*Clostridium* sensu stricto ^1^	1.88 ^a^	0.68 ^ab^	1.46 ^a^	0.48 ^a^	10.57 ^a^	0.06 ^c^	0.02 ^c^	0.07 ^c^	0.05 ^c^	0.63 ^bc^
*Lactobacillus*	48.59	17.11	36.16	43.75	31.31	27.88	12.68	14.1	24.25	14.22
*Megasphaera*	3.82 ^bc^	2.91 ^bc^	0.79 ^c^	2.31 ^bc^	4.04 ^abc^	3.36 ^abc^	6.81 ^a^	2.63 ^abc^	5.69 ^a^	4.25 ^ab^
*Romboutsia*	10.77	0.48	1.84	0.75	0.88	0.22	0.25	0.01	0.04	0.08
*Streptococcus*	1.93 ^b^	56.79 ^a^	17.83 ^ab^	15.63 ^b^	17.96 ^ab^	8.66 ^b^	14.94 ^ab^	13.9 ^ab^	7.34 ^b^	4.75 ^b^
*Turicibacter*	12.01 ^a^	0.56 ^ab^	9.56 ^a^	2.16 ^a^	6.59 ^a^	0.18 ^bc^	0.03 ^c^	0.17 ^c^	0.14 ^c^	0.41 ^c^
Proteobacteria										
*Escherichia–Shigella*	3.71 ^ab^	0.3 ^ab^	13.74 ^a^	15.61 ^ab^	12.44 ^ab^	0.43 ^ab^	0.07 ^b^	1.2 ^ab^	1.69 ^ab^	2.27 ^ab^

^1^ BB1: botanical blend 1; BB2: botanical blend 2. Each treatment had 11–12 observations. ^a–c^ Means without a common superscript are different (*p* < 0.05).

**Table 8 microorganisms-11-00320-t008:** Serum and ileal mucosa metabolites that differed among the dietary treatment groups.

Metabolite	Fold Change ^1^	VIP ^2^	FDR ^3^
CON− ^4^ vs. CON+ ^5^, serum d 4, post-inoculation (PI)			
oleic acid	0.32	1.33	0.048
arachidonic acid	0.36	1.61	0.121
lauric acid	0.39	1.53	0.133
methionine	2.01	1.59	0.121
malic acid	2.32	1.68	0.100
galactonic acid	2.48	1.49	0.145
pinitol	3.15	1.89	0.048
CON+ vs. 100 ppm BB2 ^6^, serum, d 4 PI			
pinitol	0.47	2.09	0.180
CON− vs. CON+, serum, d 21 PI			
guanosine	0.39	1.79	0.114
methionine	0.45	1.96	0.067
mannose	2.49	2.27	0.004
CON+ vs. 100 ppm BB2, serum, d 21 PI			
cholesterol	0.36	2.02	0.190
aminomalonic acid	0.44	1.92	0.190
heptanoic acid	2.25	2.06	0.190
CON+ vs. 50 ppm BB2, ileal mucosa, d 5 PI			
asparagine	0.49	2.35	0.160

^1^ Fold change values less than one indicate that the differential metabolites were reduced in CON– compared with CON+, or CON+ compared with 50 ppm BB2, or CON+ compared with 100 ppm BB2, respectively. ^2^ VIP = variable importance in projection. ^3^ FDR = false discovery rate. ^4^ CON– = negative control; basal nursery experimental diet, with ETEC challenge. ^5^ CON+ = positive control; basal nursery experimental diet, without ETEC challenge. ^6^ BB2 = botanical blend 2.

## Data Availability

The gut microbiota data presented in the study are deposited in the SRA database repository accession number: PRJNA909269.

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
