# Peer review of "Dietary Supplementation with Botanical Blends Modified Intestinal Microbiota and Metabolomics of Weaned Pigs Experimentally Infected with Enterotoxigenic Escherichia coli"

_microorganisms, 2023, doi:10.3390/microorganisms11020320_

Round 1

Author Response

Question 1: I may not get about what is rational for studying formulation of BB1 and BB2? Maybe that authors can describe more about this in the introduction part of the manuscript.

RESPONSE: The rationale for further investigating BB1 and BB2 on gut microbiota and metabolomics profiles of serum and intestinal mucosa was described in the introduction. More information was added to the introduction and discussion sections in the revised manuscript.

Question 2: Authors mentioned both BB1 and BB2 could reduce (P < 0.05) the relative abundance of Blautia but increase (P < 0.05) the relative abundance of Escherichia-Shigella on d 5 PI compared with CON-. However, as we know, Escherichia-Shigella is a kind of pathogen causing diseases. So, from this, we cannot see the benefits of BB1 and BB2 to pigs. How can you explain this?

RESPONSE: Unfortunately, we don’t have a clear explanation on the increased abundance of Escherichia-Shigella in fecal samples due to the limits of 16S rRNA sequencing. The sequence was not characterized the gut microbiota to the species level and the bacterial biomass was not measured. The limitation of this experiment was added to the conclusion as well.

Question 3: since authors mentioned BB1 and BB2 may influence inflammation in the discussion part. So, I am curious that whether authors detect inflammation markers in pigs treated with BB1 and BB2 and controls? Thanks

RESPONSE: The systemic inflammation markers were analyzed and published in Wong et al. (2022). The related information was also added the revised manuscript (L528-530).

Reviewer 2 Report

Dear authors, my major concern is the lack of interest to the readers, I suggest adding more emphasis on highlight the new of your research. In the attached file my suggestions.

Author Response

Thank you for your great revision!

Each comment was responded in the attached file. 

Please refer to the revised manuscript for the detailed revisions. 

Again, your review is highly appreciated!

Round 2

Reviewer 1 Report

Dear Editor,

I am satisfied about revised manucript by authors. Thank you so much.

Br,

Yin

Author Response

Dear reviewer,

Thank you very much for your time and effort to review this manuscript.

We truly appreciate your considertation.

Sincerely,

Yanhong

Reviewer 2 Report

Dear authors, thank you for agreeing with my reviews, I believe that the paper improved a lot. I have just one comment that was not addressed. Please see the attached file.

Author Response

Dear reviewer,

We made a slight change in the sentence by only focusing on pigs. The recommended citations are great and related to feed additives, but not relevant to pigs or non-ruminants. We highly appreciate your suggestion. But we would like to keep the current version.

We truly appreciate your time and effort for reviewing and improving the overall quality of this manuscript.

Thank you again for your consideration.

Sincerely,

Yanhong